# Enhancement of 3D Printability by FDM and Electrical Conductivity of PLA/MWCNT Filaments Using Lignin as Bio-Dispersant

**DOI:** 10.3390/polym15040999

**Published:** 2023-02-17

**Authors:** Silvia Lage-Rivera, Ana Ares-Pernas, Juan Carlos Becerra Permuy, Anne Gosset, María-José Abad

**Affiliations:** 1Universidade da Coruña, Campus Industrial de Ferrol, CITENI-Grupo de Polímeros, Campus de Esteiro, 15403 Ferrol, A Coruña, Spain; 2Universidade da Coruña, Campus Industrial de Ferrol, CITENI, Campus de Esteiro, 15403 Ferrol, Spain

**Keywords:** FDM 3D printing, electrically conductive filaments, PLA/MWCNT, polymer composite, lignin, biopolymers, bio-dispersants

## Abstract

To increase the applications of FDM (fusion deposition modeling) 3D printing in electronics, it is necessary to develop new filaments with good electrical properties and suitable processability. In this work, polymer composites filament-shaped with superior electrical performance based on polylactic acid (PLA) carbon nanotubes and lignin blends have been studied by combining solution mixing and melt blending. The results showed that composites achieve electrical percolation from 5 wt.% of nanotubes, with high electrical conductivity. Moreover, the introduction of a plasticizing additive, lignin, improved the printability of the material while increasing its electrical conductivity (from (1.5 ± 0.9)·10^−7^ S·cm^−1^ to (1.4 ± 0.9)·10^−1^ S cm^−1^ with 5 wt.% carbon nanotubes and 1 wt.% lignin) maintaining the mechanical properties of composite without additive. To validate lignin performance, its effect on PLA/MWCNT was compare with polyethylene glycol. PEG is a well-known commercial additive, and its use as dispersant and plasticizer in PLA/MWCNT composites has been proven in bibliography. PLA/MWCNT composites display easier processability by 3D printing and more adhesion between the printed layers with lignin than with PEG. In addition, the polyethylene glycol produces a plasticizing effect in the PLA matrix reducing the composite stiffness. Finally, an interactive electronic prototype was 3D printed to assess the printability of the new conducting filaments with 5 wt.% of MWCNT.

## 1. Introduction

The additive manufacturing (AM) [1,2] industrial revolution is very promising regarding the fabrication of electronic devices [3] due to its many advantages (no waste production, final product adaptability, low cost, etc.). One of the most important [4,5,6,7] AM techniques is fusion deposition modelling (FDM) [8], basing its operation on the melting of a thermoplastic material [9].

Nowadays there are commercially available thermoplastic filaments to easily feed the 3D printer (acrylonitrile butadiene styrene (ABS), polyethylene terephthalate glycol (PETG), or polylactic acid (PLA). Moreover, researchers are looking for novel filaments from other thermoplastic polymers such as PVC [10] or PLA-TPU [11]. However, these materials possess lack functional properties (such as electrical conductivity) to be used in the fabrication of such electronic devices. Readily accessible filaments with functional properties such as electrical conductivity exist on the market but they are still scarce and expensive, so FDM applications in the electronic field are limited [12].

The development of conducting polymer composites (CPC), where a conductive filler is incorporated into a thermoplastic matrix, is a good alternative for the production of printable conductive filaments [8,12,13], whenever materials should display suitable physical properties (electric, rheological, thermal and mechanical properties) to the 3D printing process. Previously published studies display the use of conductive polymer composites in energy storage devices [6,14], such as electrodes [15,16,17] or electrolytes [18], in electromagnetic interference shielding [19,20], electronics industry [21] and biomedical application [22,23].

The use of PLA in filaments production for 3D printing is not new. PLA [24], is an amorphous thermoplastic aliphatic polyester, and one of the most known biopolymers in the AM industry [25]. Due to this precedence, it is biodegradable and compostable under certain conditions. Its low thermal expansion coefficient gives rise to high dimensional accuracy in the 3D printed pieces [7]. It has good mechanical properties (stiff and strong), high thermal and light resistance and it can develop piezoelectricity [26] when it is film-form. However, PLA has some drawbacks such as its high fragility or its electrical insulating properties (σ ≈ 10^−18^ S·cm^−1^ [24]). The addition of electrically conductive nanofillers can improve the electrical conductivity of PLA, forming a conductive path through the polymer matrix. Several researchers have already focused their investigations on carbonaceous nanofillers [24] such as carbon nanofibers (CNF) [27], carbon black [25], multiwalled carbon nanotubes (MWCNT) [28,29,30], graphite [31] or graphene [32,33], always looking for a proper filler dispersion through the polymer matrix, key point to obtain low electrical threshold and the maximum conductivity. Among the carbon-based nanofillers available, MWCNT has excellent electrical properties and high aspect ratio, obtaining percolation threshold with a low amount of nanofiller. For these reasons, some authors [13,34,35] propose their use as mechanical and thermal reinforcement into PLA.

However, the improvement of the electrical conductivity of MWCNT/PLA compounds comes at the expense of viscosity increase and lower ductility, which makes the material useless in FDM. These problems can be solved using commercial plasticizers such as polyethylene glycol (PEG) [36,37], oligomeric PLA [38], Struktol [39], triethyl citrate (TEC) or acetyl tributyl citrate (ATCB). In addition, current studies are incorporating biomaterials as natural additives [40] are more environmentally friendly. One of these materials is lignin [41], which is presented in great abundance in nature since it is the second most common biopolymer on Earth, although its use as a bio-additive in conducting polymer composites for 3D printing has not been studied.

From the review of the most significant data published so far, shown in Table 1, several conclusions can be drawn: the electrical conductivity achieved by the composites is low and the processing of the composite is complex and difficult to scale at the industrial level. In the current work, we propose a combination of two methods (solution and melt mixing) to obtain a highly electrically conductive material with easy printability. This new composite will allow all the advantages of the technique to be exploited in the fabrication of new products for electronic applications.

The main objective of this paper is to obtain a new conductive filament for 3D printing, based on a PLA matrix, a conductive carbon-based nanofiller (MWCNT), and a bio-additive (lignin). The obtaining of the composite was optimized by a combination of solvent-casting and melt-mixing, obtaining highly conductive polymer composites in a safe way. From an in-depth study of the rheological, thermal, and mechanical properties of composites, the most balanced formulation was selected. Finally, the incorporation of lignin into the formulation was studied to improve its final properties and facilitate its application in a 3D printer. The use of lignin as a bio-plasticizer was compared with PEG, a synthetic and well-known additive for the PLA/MWCNT composite. Last, an interactive demonstrator with LED lights was printed using the developed composite, showing the material printability and its excellent electrical conductivity.

## 2. Materials and Methods

PLA 4043D (supplied by Nature Works, Minnetonka, USA) was employed as matrix. It has 1.24 g∙cm^−3^ of density, its melting point is 151 °C and its flow index is 6 g/10 min (210 °C, 2.16 kg). MWCNT (NC7000TM, Nanocyl, Sambreville, Belgium) have an average diameter of 9.5 nm, a length of 1.5 mm, a surface area of 250–300 m^2^∙g^−1^ and they possess an electrical conductivity of 104 S∙cm^−1^ accordingly to the supplier. The used solvents were DCM (Merck, Sigma Aldrich, Madrid, Spain) with purity > 99.9% and a density of 1.33 g∙cm^−3^ and acetone, synthesis grade (Scharlau, Barcelona, Spain), with a density of 0.79 g∙cm^−3^. Lignin was obtained from Betula alba dark, a typical forest tree in Spain’s northwest by Organosolv fractionation (Acetosolv [44,45,46]). The obtained lignin was extracted from acetic acid solution by water precipitation and lyophilized before use. Its characterization was performed by gel permeation chromatography, obtaining Mw = 2800, Mn = 1706 and polydispersity = 1.6 (average of 2 replicates). Last, polyethylene glycol 2000 for synthesis (PEG) with a density of 1.21 g∙cm^−3^ was supplied by Sigma-Aldrich, Madrid, Spain. All the materials were used as received except the PLA, which was previously dried 24 h at 60 °C.

### 2.1. Preparation of the Composites

The novel composites were obtained by a combination of solvent-casting and melt-mixing methods. The complete obtaining flowchart is exhibited in Figure 1. First, a PLA/MWCNT Masterbatch was prepared by solvent-casting method (following the reported literature [20], 20wt.% MWCNT/80wt.% PLA, named M20). This step enables a proper dispersion of the MWCNT into the PLA matrix and it facilitated the operation with MWCNT in the extruder. The PLA/MWCNT solution was evaporated in an extractor-hood with forced ventilation and then cut uniformly into small pieces with a guillotine.

Then, the PLA/MWCNT, PLA/MWCNT/lignin, and PLA/MWCNT/PEG composites were melt-blending with a Minilab Haake Rheomex CTW5 (Thermo Scientific, Massachusetts, USA) by adding the correct amount of M20, PLA bulk and lignin or PEG at 210 °C and 40 rpm, picking up the mixed material after 5 min of blending. Different samples (coins and bone dog samples) of each PLA/MWCNT composite were injection-molded in a Haake MiniJet Pro (Thermo Scientific) at 210 °C (mold at 60 °C), a pressure of 800 bar for 6 s and post-pressure of 500 bar for 3 s to evaluate their physical properties.

To study different physical properties of the new composites, several composites were obtained. Each sample was labeled as PxC, PxCyL or PxCzG, where x is the percentage of MWCNT (in the range between 2 and 15 wt.%) in the sample, y is the percentage of lignin and z is the percentage of PEG (both additives in the range between 1 and 3%) which has been calculated taken on account of the MWCNT and the PLA as a common phase.

### 2.2. Three-Dimensional Printing Process

PxCyL and PxCzG composites were shaped into coins by 3D printing process. The selected 3d printer was a modified Creality CR-10 v2 with a pellet extruder. The printing conditions were 200 °C at the extruder, hot bed at 60 °C, nozzle of 0.8 mm, layer high of 0.3 mm, superficial ironing of 5% flow rate with 0.1 mm of separation between ironing passes, and printing speed of 10 mm∙s^−1^. The samples were printed with 100% infill, lineal pattern, and 3 external perimeters. 

### 2.3. Characterization Methods

#### 2.3.1. Scanning Electron Microscopy (SEM)

The nanocomposite morphology and MWCNT dispersion into PLA matrix were analyzed using a JEOL JSM-7200F field emission scanning electron microscope (SEM) at an accelerating voltage of 5 kV and different magnifications. Previously, samples were cryogenic-broken and sputtered with platinum and palladium. The fracture was made to evaluate the cross-sectional microstructure of the samples. At least 2 samples of each formulation were broken, prepared, and analyzed by SEM to ensure acceptable reproducibility.

#### 2.3.2. Electrical Conductivity

The electrical conductivity (σ) was measured at room temperature with a LORESTA-GP (MCP-T610, Mitsubishi Chemical, Tokio, Japan) electric analyzer by the 4-point method (gold contact wires with an inter-pin distance of 5 mm) in circular plaques prepared by injection molding and 3D printing (25 mm of diameter, 1.7 mm of thickness). The σ obtained for each composite formulation are the result of at least 12 measurements on three different samples on the top and bottom surfaces to confirm the homogeneity of the circular plaques.

#### 2.3.3. Rheology

For the viscoelastic characterization, a controlled strain rheometer (ARES, TA Instruments, New Castle, DE, USA), composed by parallel-plate geometry (25 mm diameter, 1mm gap) was used. The samples were measured in nitrogen atmosphere to avoid sample oxidation during the tests. The rheological tests were performed in the linear viscoelastic region (LVE), where the modulus is independent of the strain, to the temperature at which the composites were extruded (190 and 210 °C). LVE region was determined by a strain sweep test before testing the viscoelastic properties of the sample. Then, viscoelastic parameters were measured in the frequency range (σ) from 10^−1^ to 10^2^ rad/s. Each curve reported is an average of at least two samples.

#### 2.3.4. Tensile Test

The mechanical properties were measured by uniaxial tensile tests, performed at a crosshead speed of 2 mm∙min^−1^ at room temperature using an Instron 5566 universal test machine (Instron, Canton, MA, USA) according to UNE-EN ISO 527-2. Young’s modulus (E), stress (σ_y_) and strain (ε_y_) at the yield point, and stress (σ_B_) and strain (ε_B_) at the breakpoint and their corresponding standard deviations were obtained aiming to study the mechanical performance of the different samples. At least five specimens of each sample were tested to obtain the average value of the mechanical properties and their standard deviations.

#### 2.3.5. Differential Scanning Calorimeter (DSC)

The different samples were analyzed by differential scanning calorimetry (2010 DSC TA Instruments, New Castle, USA) under a nitrogen atmosphere to evaluate the influence of MWCNT on crystallization behavior of PLA and on the glass transition temperature of the composites. The samples (10–15 mg) were heated from 0 °C to 200 °C at a rate of 10 °C∙min^−1^ and maintained at 2 min at 200 °C to erase their thermal history. Then, they were cooled to 0 °C at a rate of 10 °C∙min^−1^ and heated again to 200 °C to measure the characteristic temperatures and enthalpies. Every result is the average of at least 2 measurements in different specimens.

#### 2.3.6. Thermogravimetric Analyses (TGA)

To evaluate the thermal stability of the composites, thermogravimetric analysis was performed using a TGA 4000 Perkin Elmer, MA, USA. The samples were heated from 50 to 700 °C at 10 °C·min^−1^ under a nitrogen atmosphere. From TGA thermograms, the degradation temperature (calculated as onset temperature in thermogram) and the residual mass at 500 °C were determined.

## 3. Results and Discussions

### 3.1. Solvent Optimization Study

First, the selection of the best solvent combination, which allows a proper dispersion of MWCNT into PLA without matrix degradation was carried out. Different PLA-dissolved samples were obtained by a combination of two organic solvents: DCM (which presents a high affinity towards dissolution) and acetone (which presents a poor affinity towards dissolution). The two solvents were selected because they possess a relatively low boiling point (40 and 56 °C, respectively, allowing fast evaporation), they are both able to dissolve the PLA and they are not carcinogenic. Table 2 displays the different DCM/acetone combinations, with a 1:10 PLA/solvent proportion, studied to compare the physical properties of PLA samples prepared by the same obtained method of the composites one (solvent casting with different solvents, extrusion, and injection). In addition, PLA samples were injected (S0) and extruded, and then injected (S1) as a reference.

The first step to choosing the best solvent was performing the viscoelastic characterization. The tests were run at 190 °C (matching the processing temperature). The complex viscosity η* dependence on frequency is collected in log-log plots in Figure 2A. The samples exhibit Newtonian behavior and only a small frequency dependence at high frequencies as was expected in pure polymers (Figure 2A). If the samples are compared between them, a significant drop between S0 and S1 (the non-dissolved samples) η* is displayed, this is produced by the break of the biopolymer chains during the extrusion process. Comparing S2 and S3 samples it seems clear that DCM is much more aggressive to PLA than acetone. The shortening of polymer chains with solvent produces an important diminution in viscosity values (S3). Nevertheless, despite S2 develops good results, the solution time is very high so, this solution method is not the most appropriate. Adding a little amount of acetone to the DCM solvent (S4) it is possible to obtain a minor degradation of PLA chains and reduce the solution time considerably with respect to the dissolved sample with only acetone. The addition of a high percentage of acetone (S5) produces an important degradation in the sample reflected in very low viscosity. Moreover, in Figure 2B all the samples show a like-liquid solid (G″ > G′) response as was expected for a pure polymer. Considering these observations, the most appropriate solvent combination would be 4:1 DCM:acetone, that is S4 sample.

From tensile tests, the mechanical parameters were obtained and are displayed in Table 2. Comparing the solved PLA samples, the material stiffness remained constant within the deviations. Only the S4 sample showed a slight decrease in Young’s modulus with respect to the other ones. Yet the main effect of PLA degradation in mechanical properties was observed in a severe decrease in the material ductility due to the shortening of biopolymer macromolecules after the solvent process. In addition, the increase in tensile strength at the yield point of S1, S2, S3, and S5 is a result of the fact that the samples broke right after reaching the fluency yield point (see Appendix A). Considering the mechanical results, it seems that the S4 sample suffers less degradation during the dissolution process. This fact matches well with the rheological results. Therefore, the procedure followed to prepare the S4 sample was chosen for the solvent casting process [17,47] from now on, and all the composites of this study were made with it.

### 3.2. Optimization of MWCNT Content

In order to calculate the electrical percolation threshold [48], MWCNT/PLA composites with different carbon nanotube amounts (Table 3) were prepared following the procedure explained in “Preparation of the composites” section.

The microstructure of PLA/MWCNT composites was studied by SEM. Some MWCNT agglomerates are displayed in micrographs (see red arrows in Figure 3). Their number and size increased with the nanofiller amount, as can be seen in Figure 3. In addition, all micrographs show MWCNT agglomerates and smooth areas with individual and dispersed MWCNT, which connect the different agglomerates giving rise to an electrically conductive network (inset images in Figure 3) [49]. This effect is clearly shown in Figure 3C. The P5C composite developed well-connected agglomerates with individual MWCNT. When the filler amount increases, the differences between the agglomerates and its smooth zone are not so easily discernible (see insets in Figure 3D). This could be evidence that the electrical percolation has already occurred.

To corroborate the morphological findings, the percolation threshold value was determined from the measurement of electrical conductivity of PLA/MWCNT composites (Appendix A) following the equation below, where σ_0_ is the effective conductivity of the MWCNT within the PLA, ρ and $\rho_{c}$ are the real and the critical volumetric concentration of nanofiller, and t is the critical exponent which depends on the dimensions of the conducting network:(1)$$\sigma =\sigma_{0}\;{\left(\rho -\rho_{c}\right)}^{t}$$

Figure 4 represents the electrical conductivity values of composites as a function of MWCNT content. Graphically, the electrical percolation threshold is reached between 5–5.5 wt.% MWCNT, since the electrical conductivity suffers an increment of 6 magnitude orders. Moreover, between 6–15 wt.% MWCNT the electrical conductivity value remained practically stable at 1.8 ± 0.2 S∙cm^−1^. The inset in the figure shows the mathematical linear adjustment to obtain the experimental value for the electrical threshold value ($\rho_{c}$). The best linear fit (R^2^ > 0.99) was found for $\rho_{c}=4$8.3 vol% MWCNT (5.0 wt.%), matching with the morphological results. Last, the critical exponent is 1.2, which falls in the double dimensionality of the conductive network as previously reported in the literature (between 1.1 and 1.3) [50].

Next, rheological measurements were performed to characterize the percolation state of the nanofillers and their dispersion in the PLA matrix. Additionally, the rheological characterization of the composite is important to guess the behavior of the material in the printing process. Viscosity affects the proper extrusion of the material without die swell effect and filament buckling and the viscoelastic properties give rise to a proper layer adhesion [51]. First, complex viscosity was studied as a function of both frequency and nanofillers addition (Figure 5A). MWCNT incorporation caused a considerable change in η* values, increasing in comparison with the PLA ones (S4). This tendency was more pronounced at low frequencies (Appendix A) because the relaxation of the polymer chains was restricted by the presence of the MWCNT [52]. Furthermore, every PxC sample showed the typical shear thinning behavior as is expected for filled composites. In addition, there was a considerable increment (several magnitude orders) between P2C, P4C, and P6C viscosities at low and medium frequencies. Following the reported studies [52], the rheological threshold was visible on viscosity curves (Figure 5A) looking at the change from Newtonian to time-dependent behavior or in the G′ graphic noticing the appearance of a plateau (Figure 5B). This response, observed from the P2C composite, was attributed to the viscoelastic transition from liquid-like to solid-like behavior.

The loss modulus replicated the same behavior as G′ modulus. The S4 sample shows liquid-like behavior (G″ > G′). A crossover point is visible for P2C at a high frequency of around 70 rad∙s^−1^ changing the G″ > G′ behavior before this frequency to G′ > G″ afterward. The rest of the samples show a solid behavior (G′ > G″) during the whole frequency range indicating that the percolated net is totally formed [53]. The conductive network increased the number of interfaces through nanotubes, increasing the elastic response (solid-like behavior) together with an increase in energy dissipation component (G″), which explains the increase in both moduli.

The liquid–solid transition can be observed too at the van Gurp–Palmen graph [54] (Figure 5C) which plots the phase angles (δ) against the complex moduli (G*). S4 values were near to 90 ° (the material was totally relaxed), but MWCNT addition deviated δ and the gap increased with the percentage of nanofiller. This effect showed the formation of a percolated structure in the melted sample.

To conclude the viscoelastic characterization, the rheological threshold was calculated, similar to the electrical one, from the adjustment of G´ values, at low frequency (0.1 rad·s^−1^) to the following equation:(2)$$G^{\prime }={G^{\prime }}_{0}\;{\left(\rho -\rho_{c},G^{\prime }\right)}^{\mathrm{tc},G^{\prime }}$$

The best fitting (R^2^ > 0.98) is shown in the inset of the Figure 5D with $\rho_{c}$,G′ = 33.6 vol% and tc,G′ = 0.55 ± 0.03. The rheological threshold is lower than the electrical one. According to some authors [52,55], the rheological percolation is reached when the average distance is from 10 to 100 nm, so the MWCNT are not in direct contact with each other. However, electrical percolation is achieved when a conductive path is formed throughout the material. Therefore, when the nanotubes are in direct contact with each other (direct conduction mechanism) or when the distance between nanotube and nanotube is less than 5 nm (electron hopping–tunneling mechanism). This would clearly explain the differences between the electrical and rheological threshold values.

Upcoming, MWCNT influence on the tensile strength of PLA was studied and collected in Table 3. Increased stiffness is associated with increased composite brittleness as a function of the amount of MWCNT [21,56]. Consequently, the stress at the breakpoint of composites is higher than the reference sample (S4) because the breakpoint happens before a yield (Appendix A), giving rise to ductility loss. This brittle behavior can be overcome by adding a proper plasticizer [57].

From TGA analysis, the effect of nanotubes on the degradation temperature of PLA was analyzed. In addition, the residual mass in thermograms allowed the measurement of the real MWCNT amount incorporated in composites (Table 3). Within the deviations of the technique, the residual mass at 500 °C agrees with the theoretical nanotube content in PLA/MWCNT composites. As it has previously been reported in the literature [58], the addition of MWCNT enhances the thermal stability of PLA compared to the reference sample (S4), increasing the value of thermal degradation temperature, measured as the onset in TGA thermograms (Tonset). However, the increase is not linear to the nanotube content. After the electrical threshold, the Tonset values drop with the nanofiller amount. Probably, the high shear during PLA extrusion with a high MWCNT amount causes thermal degradation in the PLA matrix and, consequently, decreases the thermal stability of PLA/MWCNT composites.

The influence of nanofiller amount on the melting and crystallization behavior of PLA was analyzed by DSC (Appendix A). First, the glass transition temperature (Tg) of PLA remained constant with MWCNT addition. Second, PLA possesses a low crystallization rate during the cooling (due to its asymmetrical structure of chains [36]), but with the filler incorporation, PLA showed a notorious cold crystallization peak. These changes can be attributed to the nucleating effect provoked by MWCNT. This heterogeneous nucleation results in small variations in the cold crystallization temperature (T_cc_) and the melting temperature (T_m_) as a function of MWCNT content. Both cold crystallization and melting enthalpies (see Appendix A) are similar for P2C, P4C, and P6C, proving that they are fundamentally amorphous and only crystallize during the heating scan. However, P8C and P10C show almost no crystallization and melting peaks. Probably, the high content of nanofillers restrains the motion and arrangement of the molecular chains, hindering the growth of crystallization nuclei [59,60]. This behavior agrees with morphological analysis by SEM, rheology, and electrical conductivity results. In addition, the crystallization and melting enthalpies of each sample are similar within the deviations, showing that the PLA matrix is mainly amorphous after cooling. This fact is beneficial for the 3D printing process.

### 3.3. Optimization of Additives Content

To enhance the PLA/MWCNT electrical conductivity and obtain a more flexible and ductile material, changing the nanofiller dispersion and distribution through the polymer matrix, different amounts of lignin [61] and PEG [37] were added to composites. The new samples were formulated from PC5 composite, since 5 wt.% was the critical amount to achieve the electrical percolation. The new formulations are summarized in Table 4. With the aim of studying the viability of lignin as a bio-additive in P5C, as well as comparing its performance with a commercial additive PxCyL and PxCzG composites were 3D printed into coin shape samples, which were subjected to SEM, rheology, and electrical conductivity tests (Appendix A). Moreover, dog bone shaped specimens were injected (210 °C) to measure the tensile properties.

The influence of lignin and PEG in PLA/MWCNT morphology was studied by SEM. The micrographs (Figure 6) display a general view (just 40× amplification) of each sample, as well as some magnifications (5000× amplification) to see the distribution of the nanofiller within the different composites. About P5CyL specimens (Figure 6A–C), MWCNT agglomerates can be appreciated in the three samples. Within these agglomerates (inset on the left) there is a perfectly interconnected network, which provides a conductive path. In the case of P5C1L, its surface is uniform, while in samples with 2 and 3 wt.% lignin agglomerations are observed. In addition, focusing on areas that do not have MWCNT agglomerates, small groups of nanofillers (MWCNT bundles), single carbon nanotubes, and small lignin agglomerates are shown. These lignin aggregations grow as bio-additive content increases in formulations. In addition, the adhesion between layers improves with lignin since no interfaces are seen in the micrographs.

Furthermore, the micrographs of Figure 6D–F show the composite structure modified with PEG. Although the microstructure may appear similar to composites with lignin, the micrographs show a more heterogeneous surface, and the layer interfaces are more visible. This effect can be attributed to the immiscibility between PEG and PLA. Although the dispersing effect of PEG on nanotubes in PLA matrix has already been proven [36], these data display that lignin is acting as a bio-dispersant of the nanofiller as well as helping the 3D printing process.

Next, the viscoelastic behavior of the composites was analyzed at 210 °C (Figure 7) on 3D printed coin-shaped samples from the polymer composites with lignin and PEG. Both storage and loss modulus dependence with the plasticizer addition were studied (Figure 7A,B), to check the material viability to be processed by 3D printing [51]. Both lignin and PEG additives should cause a plasticizer effect, disrupting the interaction between PLA molecular chains improving their mobility, reducing the energy stored during elastic deformations and the energy losses during viscous deformations, causing consequently, an important diminution in moduli with the additive increase.

In respect of P5zL samples, both PLA and lignin are polar and therefore they are miscible between them. For that reason, the plasticizer effect of lignin in PLA is little and the changes in the viscoelastic properties are not very noticeable when lignin content is small (1 wt.%). However, P5C3L shows a decrease in both moduli with respect to P5C. Moreover, P5C3L has a lignin excess, and both moduli increase even higher than P5C. This behavior can be attributed to a lignin excess, which concords with lignin agglomerates observed in SEM micrographs of P5C3L. Regarding P5zG samples, the modulus was affected by PEG incorporation due to the dispersion effect induced on the MWCNT, as can be observed in the morphological study. The PEG caused an expected plasticizer in the composite in the samples with 2 and 3 wt.% of additives. This plasticizer effect causes an important diminution in moduli with the PEG increase (2 and 3 wt.%) phase separation between PLA and PEG as observed in SEM micrographs.

Next, the tensile tests were performed to analyze the influence of lignin and PEG on the mechanical properties of PLA/MWCNT composites. The data collected in Table 4, show that five mechanical parameters remained constant (taking into account standard deviations) with lignin addition. The bio-additive works as a nanofiller‘s dispersant without plasticizing effects in the PLA matrix. On the contrary, PEG produces a decrease in elastic modulus and increases in elongation at the breakpoint of the P5C composite (see Appendix A). The dispersant effect of PEG is associated with a plasticizing effect in biopolymers enhanced by the phase separation between them, as was seen in SEM micrographs.

Next, the influence of lignin and PEG in the electrical conductivity of the composite was studied. Both additives increased the electrical conductivity of PLA/MWCNT in six magnitude orders due to the dispersant effect that lignin and PEG caused in the composite. For the same additive content, there is no substantial difference between lignin and PEG. It seems evident that the inclusion of the dispersant additive allows the formation of a stronger conductive network with MWCNT agglomerates well inter-connected.

Considering all the data, 1 wt.% of lignin has been selected as the optimum amount of bio-additive to improve PLA/MWCNT electrical conductivity as well as printing process.

### 3.4. Electrically Conductive 3D Printed Prototype

Herein, to demonstrate the superior conductive nature of this novel material, an interactive prototype with an electrically conductive filament composed of PLA and 5 wt.%MWCNT (electrical conductivity of 2.1∙10^−1^ S∙cm^−1^) was printed in a modified dual extruder Mendel Max XL v4 (Makergal, Santiago de Compostela, Spain). The prototype was processed at 210 °C with a 0.4 mm nozzle and 0.3 mm of layer height. To enhance the adhesion of the first layer of the piece, the bed temperature was set to 60 °C. The printing process was monitored in real time in order to ensure non-destructive quality control of the final piece. Concretely, the material continuity and internal density were essential to obtaining the targeted electrical conductivity, discarding the pieces that did not meet the standards. In addition, the infill option was configured with a 100% density and rectilinear pattern. Last, each part of the prototype was subjected to top superficial ironing with a 15% of flow rate and 0.1 mm of spacing between ironing passes, obtaining a smoother and better looking final product. The device has three LEDs of different colors and a selector piece (also printed in 3D) that closes the electronic circuit of each LED turning on each of them. The four states of the prototype are collected in Figure 8. Each LED requires a different voltage and intensity; therefore, the length of every individual circuit is different, and it has been calculated taking into account the resistivity of the conductive filament and the LED’s necessities. To activate the three circuits, 27V is needed.

## 4. Conclusions

In the current work, a conductive PLA/MWCNT composite suifigure for FDM 3D printing was obtained. MWCNTs were pre-dispersed in the PLA matrix by solvent-casting method (with an optimized solvent combination 4:1 of DCM:acetone), facilitating a first dispersion of them throughout the polymer matrix. The composite was obtained in a two-step process by solvent casting and melt-mixing, giving rise to an optimized filament with superior electrical conductivity. In addition, the procedure is easily scalable, allowing the obtaining of a big quantity of material at a reasonable cost. The electrical threshold of PLA/MWCNT composite was found at 5 wt.% (48.3 vol%) of nanofiller with an electrical conductivity of (2.8 ± 0.1)∙10^−7^ S∙cm^−1^. The rheological threshold was found below the electrical one, with 33.6 vol% MWCNT.

The incorporation of a 1wt% of lignin in PLA/MWCNT (5 wt.% MWCNT) produced the strengthening of the conductive network, increasing the electrical conductivity in six magnitude orders, without plasticizer effect in PLA matrix. In addition, the processability of the PLA/MWCNT with 5 wt.% of nanofiller by 3D printing was improved and the SEM micrographs show a good adhesion between printed layers. The electrical conductivity of conducting polymer composites was also increased with the PEG addition at the expense of plasticizing the polymer matrix. No improvement in the printing process was observed when PEG was added, probably due to its immiscibility with polylactic acid.

To sum up, we have obtained a conductive (1.7∙10^−1^ S∙cm^−1^) and easily printable filament which contains only 5 wt.% of MWCNT thanks to the effect of the lignin (P5C1L), a bio-based material. The new material developed suitable rheological properties to be printable while being highly electrically conductive.

## Figures and Tables

**Figure 1 polymers-15-00999-f001:**
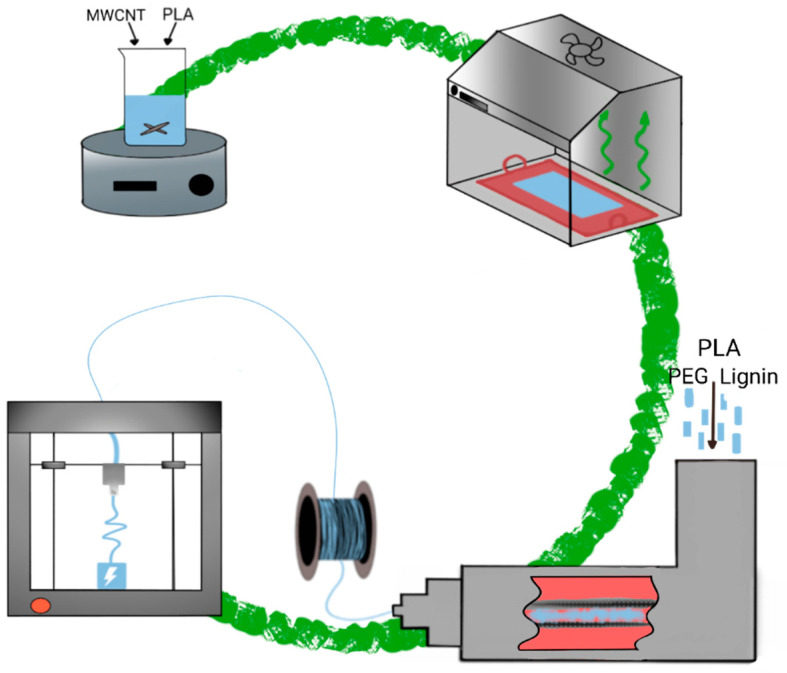
Flowchart of the polymer composites obtaining process: solvent casting (with the extractor hood), melt mixing, and 3D printing.

**Figure 2 polymers-15-00999-f002:**
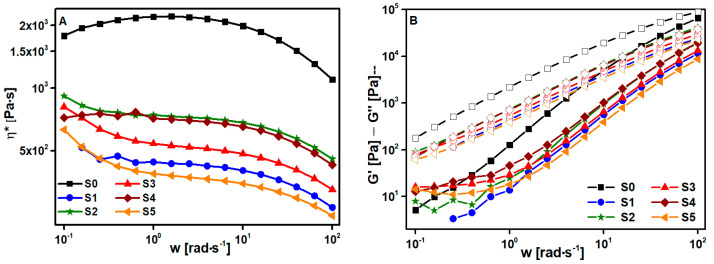
(**A**) Complex viscosity (η*) and (**B**) crossing points of the solvent optimization study, in which the closed marks are G′ and the open ones are G″.

**Figure 3 polymers-15-00999-f003:**
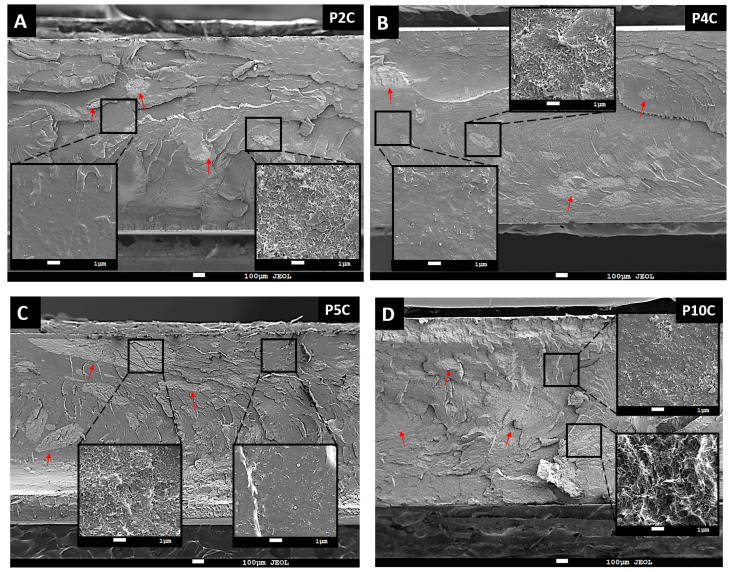
SEM micrographs of injected samples of PLA with (**A**) 2, (**B**) 4, (**C**) 5 and (**D**) 10 percent of MWCNT, amplitude ×40 and inset ×5000.

**Figure 4 polymers-15-00999-f004:**
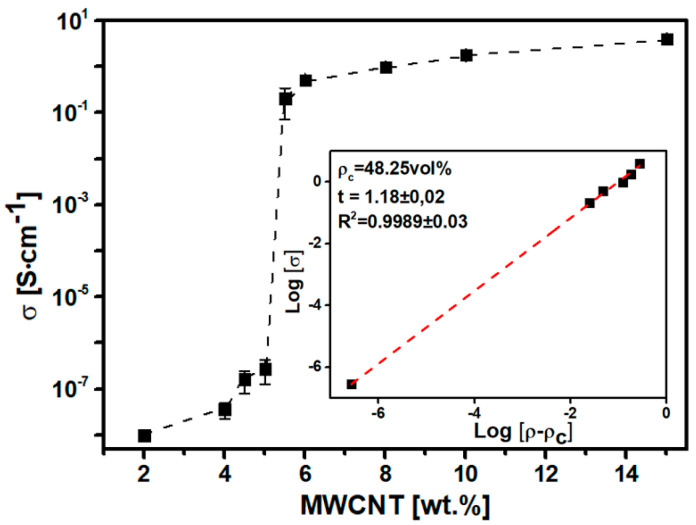
Electrical threshold of PLA/MWCNT composite and linear fit adjustment (inset).

**Figure 5 polymers-15-00999-f005:**
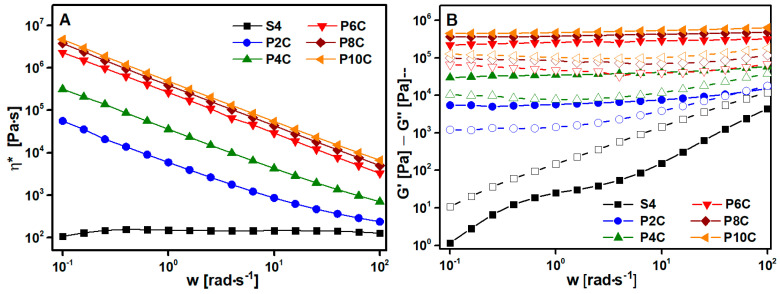
(**A**) Complex viscosity (η*) as frequency function, (**B**) frequency sweep test of dynamic storage and loss modulus, in which the closed marks are G′ and the open ones are G″, (**C**) van Gurp–Palmen graphic and (**D**) rheological threshold for composites with different MWCNT concentration and its linear fit adjustment (inset).

**Figure 6 polymers-15-00999-f006:**
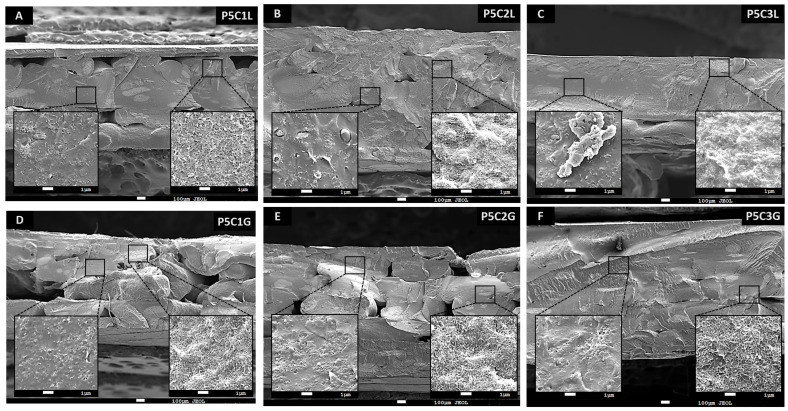
SEM micrographs of 3D printed samples of PLA/MWCNT with (**A**) 1, (**B**) 2, and (**C**) 3 percent of lignin, (**D**) 1 (**E**) 2 (**F**) 3 percent of PEG.

**Figure 7 polymers-15-00999-f007:**
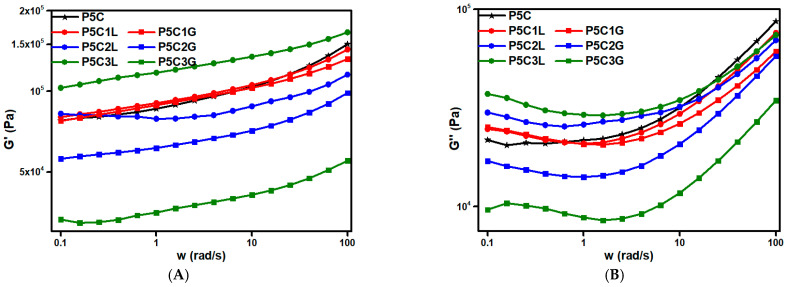
Storage modulus (G′) dependence with (**A**) lignin and (**B**) PEG addition and loss modulus dependence with (**A**) lignin and (**B**) PEG addition.

**Figure 8 polymers-15-00999-f008:**
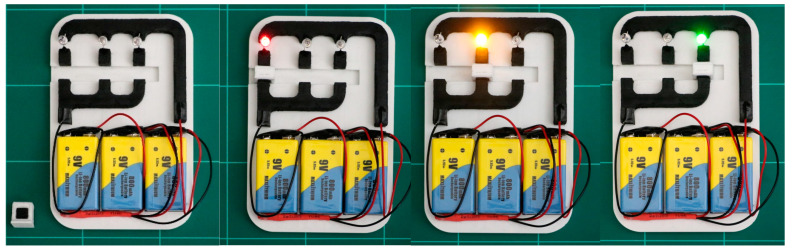
Three-dimensional printed electronic prototype with conductive (around 5 wt.% MWCNT) and non-conductive PLA-based filament.

**Table 1 polymers-15-00999-t001:** State of art historical review of reported studies with conductive PLA materials.

Materials	Filler [%]	σ [S∙cm^−1^]	Obtaining Method	Disadvantages	Ref
Graphene acting as nanofiller					
PLA/graphene	2.75	2.58 × 10^−4^	Polymerization	Lack of scalability, low electrical conductivity	[32]
PLA + TPU/reduced graphene	9	10^−5^	Solvent casting + melt mixing	Low electrical conductivity	[33]
PLA/SWCNT/graphite	1	1.25 × 10^−6^	Melt mixing	Low electrical conductivity	[31]
Carbon nanotubes acting as nanofiller					
PLA/MWCNT/Carbon black	1.5 + 1.5	9.63 × 10^−2^	Melt mixing with a third component	Lack of scalability	[25]
PLA/MWCNT	1.2	2.36 × 10^−6^	Solution blending + hot compression molding	Lack of scalability, low electrical conductivity	[21]
PLA/MWCNT	5	2.52 × 10^−1^	Extrusion method with a robot	Lack of scalability	[42]
PLA/MWCNT	6	2.1 × 10^−4^	Melt mixing	Low electrical conductivity	[43]
PLA/MWCNT	8	1 × 10^−3^	Melt mixing	Low electrical conductivity	[8]
PLA/CNT	8	10^2^	Melt mixing	High nanofiller concentration	[13]

**Table 2 polymers-15-00999-t002:** Solvent optimization study samples composition and tensile strength results. (Where E refers to Young’s modulus, **σ_B_** is the stress at the breakpoint, and **ε_B_** the strain at the breakpoint).

PLA	Code	Acetone [%]	DCM [%]	E [MPa]	σ_B_ [MPa]	ε_B_ [%]
Injected	S0	0	0	1042 ± 121	40.1 ± 1.4	122.1 ± 21.4
Extruded and injected	S1	0	0	1072 ± 242	58.6 ± 3.5	14.1 ± 3.8
Solved in acetone	S2	100	0	839 ± 224	55.9 ± 2.9	16.9 ± 2.9
Solved in dichloromethane	S3	0	100	966 ± 150	57.4 ± 1.8	14.4 ± 1.6
Solved in 4:1 DCM:acetone	S4	20	80	781 ± 186	42.1 ± 4.1	68.6 ± 9.2
Solved in 3:2 DCM:acetone	S5	40	60	954 ± 111	55.3 ± 0.9	12.8 ± 0.9

**Table 3 polymers-15-00999-t003:** MWCNT optimization study samples compositions, TGA, DSC, and tensile strength test results.

PLA	MWCNT [wt.% (vol%)]	Tonset [°C]	R_500_ [%]	Tg [°C]	E [MPa]	σ_B_ [MPa]	ε_B_ [%]
S4	-	326.3	0.5	62.03	781 ± 186	42.1 ± 4.1	68.6 ± 9.2
P2C	2 (26.6)	338.2	2.8	62.3	-	-	-
P4C	4 (42.5)	337.0	4.2	62.5	-	-	-
P4.5C	4.5 (45.5)	340.7	4.5	62.4	-	-	-
P5C	5 (48.3)	334.4	5.4	62.9	1209 ± 107	53.7 ± 8.1	6.7 ± 1.5
P5.5C	5.5 (50.8)	336.3	5.7	62.2	-	-	-
P6C	6 (53.1)	328.7	6.3	62.1	-	-	-
P8C	8 (60.6)	327.7	8.1	62.4	-	-	-
P10C	10 (66.3)	328.4	10.0	62.4	1233 ± 140	57.4 ± 8.7	8.1 ± 0.8
P15C	15 (75.8)	326.2	15.3	62.4	1517 ± 183	68.1 ± 8.1	9.2 ± 1.0

**Table 4 polymers-15-00999-t004:** Lignin and PEG percentages in samples with common PLA/MWCNT (95/5 wt.%), electrical conductivity, and mechanical data.

Sample	Lignin [wt.%]	PEG [wt.%]	σ [S·cm^−1^]	E [MPa]	σ_B_ [MPa]	ε_B_ [%]
P5C1L	1	-	(1.5 ± 0.7) × 10^−1^	1114 ± 142	47.7 ± 14.6	7.8 ± 2.0
P5C2L	2	-	(0.6 ± 0.1) × 10^−1^	1084 ± 168	49.3 ± 9.7	7.8 ± 2.3
P5C3L	3	-	(0.3 ± 0.1) × 10^−1^	1101 ± 171	54.6 ± 2.5	8.9 ± 2.0
P5C1G	-	1	(1.4 ± 0.5) × 10^−1^	1462 ± 61	39.0 ± 10.7	3.1 ± 0.9
P5C2G	-	2	(1.1 ± 3.1) × 10^−1^	840 ± 164	58.6 ± 3.0	10.5 ± 0.9
P5C3G	-	3	(0.7 ± 0.4) × 10^−1^	894 ± 229	51.1 ± 6.6	11.0 ± 2.4

## Data Availability

The data presented in this study are available on request from the corresponding author.

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
