# Peer review of "Enhancement of 3D Printability by FDM and Electrical Conductivity of PLA/MWCNT Filaments Using Lignin as Bio-Dispersant"

_polymers, 2023, doi:10.3390/polym15040999_

Round 1

Reviewer 1 Report

This paper reported a 3D printed PLA/MWCNT composite material with good electrical conductivity. The influence of each component on its conductivity and printability was discussed, and the solvents and dispersants were optimized. However, there are still some problems that need to be improved.

1. When comparing S0 and S1 in section 3.1, the decrease of η is caused by the breaking of the molecular chain, so the effect of the solvent should be point out and discussed in detail.

2. Detailed testing data of electrical conductivities should be provided.

3. Figure 9 should be changed to Figure 8. There is no Figure 8 in the manuscript.

4. The conclusion needs to be simplified and highlight the most important result and contribution of this work.

Author Response

Referee 1: This paper reported a 3D printed PLA/MWCNT composite material with good electrical conductivity. The influence of each component on its conductivity and printability was discussed, and the solvents and dispersants were optimized. However, there are still some problems that need to be improved.

  1. When comparing S and S in section 3.1, the decrease of η is caused by the breaking of the molecular chain, so the effect of the solvent should be point out and discussed in detail.

The author thanks the referee for the observation. S0 and S1 samples are PLA only injected and PLA extruded and injected, respectively, these two samples have been obtained to see the effect of the extrusion as well as for their comparison with the solved samples (S2-S5). Maybe this wasn’t explained clear enough, so we introduced a new sentence to clarify this point

(Line 223) ‘If the samples are compared between them; a significant drop between S0 and S1 (the non-dissolved samples) η* is displayed’

  1. Detailed testing data of electrical conductivities should be provided.

The author thanks the referee for the positive consideration. The electrical conductivities samples from MWCNT optimization study are shown only at Figure 4, which can make it difficult to read the data. To fix this drawback, we have added the electrical conductivities of PLA/MWCNT samples with their deviation in the Table 2 of the Supplementary Information. 

  1. Figure 9 should be changed to Figure 8. There is no Figure 8 in the manuscript.

We appreciated the observation of the reviewer. This error was corrected.

  1. The conclusion needs to be simplified and highlight the most important result and contribution of this work.

We appreciated the time and effort of the reviewer. To make the conclusions more clear, we have deleted one paragraph and rewriting another one. Now the conclusions have three paragraphs:

  • One introductory which sumps up what we have made: ‘In the current work, a conductive PLA/MWCNT composite suitable for FDM 3D printing was obtained. MWCNTs were pre-dispersed in the PLA matrix by solvent-casting method (with an optimized solvent combination 4:1 of DCM:Acetone), facilitating a first dispersion of them throughout the polymer matrix. The composite was obtained in a two-step process by solvent casting and melt-mixing, giving rise to an optimized filament with superior electrical conductivity. Besides, the procedure is easily scalable, allowing the obtaining of a big quantity of material at a reasonable cost. The electrical threshold of PLA/MWCNT composite was found at 5 wt.% (48.3 vol%) of nanofiller with an electrical conductivity of (2.8±0.1)∙10-7 S∙cm-1. The rheological threshold was found below the electrical one, with 33.6 vol% MWCNT’.
  • The next with the lignin improvements: ‘The incorporation of a 1wt% of lignin in PLA/MWCNT (5wt.% MWCNT) produced the strengthening of the conductive network, increasing the electrical conductivity in 6 magnitude orders, without plasticizer effect in PLA matrix. Besides, the processability of the PLA/MWCNT with 5 wt.% of nanofiller by 3D printing was improved and the SEM micrographs show a good adhesion between printed layers. The electrical conductivity of conducting polymer composites was also increased with the PEG addition at the expense of plasticising the polymer matrix. No improvement in printing process was observed when PEG was added, probably due to its immiscibility with polylactic acid’.
  • The last one with the formulation which we have selected as the best: ‘To sum up, we have obtained a conductive (1.7∙10-1 S∙cm-1) and easier printable filament which contains only 5 wt.% of MWCNT thanks to the effect of the lignin (P5C1L), a bio-based material. The new material developed suitable rheological properties to be printable while been highly electrically conductive’.

Reviewer 2 Report

Title, Abstract and Keywords:

The title is incomplete and should be corrected. Enhancement of what properties? Physical, mechanical or both properties? The title of the work is interesting and attractive. Especially the use of lignin as a bio-dispersant. But what is the main innovation? Also, the abstract is written very briefly. In addition, the conducted tests and their results should be added quantitatively and qualitatively. Keywords can also be modified. For what purpose is polyethene glycol added? Doesn't this require surface modification or functionalization of the CNTs?

Introsuction:

The introduction is written very briefly. Also, the sources reviewed in the introduction are very few. The introduction should be completely rewritten. The introduction of FDM, pure thermoplastic and composites, should be added in the introduction. It is suggested to use these references which are for new materials such as PVC, PLA-TPU and PETG (“Development of Pure Poly Vinyl Chloride (PVC) with Excellent 3D Printability and Macro‐and Micro‐Structural Properties” --- “3D printing of PLA-TPU with different component ratios: Fracture toughness, mechanical properties, and morphology”).

Research Method and Materials:

Why is the melt mixing method not used directly in the internal mixer or extruder? It is suggested to summarize the construction parameters in a table. Composites in a table with the weight percentage of nanoparticles and polyethene glycol and lignin should be presented. How has the reproducibility of the results been checked? The print section of the samples has been forgotten. It is necessary to provide this section due to the effect of printing parameters.

Results section:

Is it mandatory to submit section 3.1? The parameters of Table 2 should be introduced. How are the mechanical properties checked? The SEM images used are raw. These images should be corrected. Add a scale bar to them. By adding text to these images, they can be used for better analysis of the results. Are these images for printed samples? The results of the physical properties of the printed samples are presented in what plane and raster angle? Has the effect of printing parameters been checked? For example, number of microholes caused by the printing strongly affect the electrical properties? Why is the CNT distribution not studied for the printed samples? How is the accuracy of electrical property results checked?

Author Response

Referee 2:

1-The title is incomplete and should be corrected. Enhancement of what properties? Physical, mechanical or both properties? The title of the work is interesting and attractive. Especially the use of lignin as a bio-dispersant. But what is the main innovation?

The author thanks the referee for the positive feedback. In our work we would like to highlight the enhancement of the electrical conductivity of PLA/MWCNT and also its printability (easiness to be printed by a fusion deposition modelling 3D-printer) thanks to the lignin effect. That is why we decided to change a bit the title, as the referee to suggested, to

‘Enhancement of 3D printability by FDM and electrical conductivity of PLA/MWCNT filaments using lignin as bio-dispersant’.

2-Also, the abstract is written very briefly.

We appreciate the commentary of the referee. The abstract has been changed accordingly to the commentaries of this reviewer but we are not able to extend it more since there is a limit of words imposed by the journal. We trust that now the abstract is more understandable thanks to the new additions.

3-In addition, the conducted tests and their results should be added quantitatively and qualitatively.

The author thanks the reference positive feedback. We believe that the important and relevant results have been included within the paper. In addition, we have checked the manuscript again and we realized that maybe the electrical conductivities of PLA/MWCNT composites were not clearly enough, since they were only visible at the Figure 4. To fix this, we have included the numerical results as well as their deviations in the Table 2 of the supplementary information

4-Keywords can also be modified.

We thanks the reviewer assessment. In order to make more accessible the article to everybody by only searching keywords in the data bases, we decided to modify the keywords, as it was suggested by the referee: ‘FDM 3D printing, electrically conductive filaments, PLA/MWCNT, polymer composite, lignin, biopolymers, bio-dispersants’.

5-For what purpose is polyethene glycol added? Doesn't this require surface modification or functionalization of the CNTs?

The author thanks the referee for the comment. In our work we add lignin to PLA/MWCNT composite to enhance its printability and electrical conductivity. Moreover, we wanted to compare its effect with a commercial additive previously tested in this type of nanocomposites. PEG was already validated by several authors as a good additive to PLA/MWCNT, since it gets the desirable printability and electrical conductivity enhancement. That is why we decided to compare the effect of lignin and PEG, obtaining samples in parallel with both additives.  To make this point clearly, both abstract and introduction have been modified:

Line 19: ‘To validate lignin performance, its effect on PLA/MWCNT was compare with polyethylene glycol one. PEG is a well-known commercial additive, and its use as dispersant and plasticizer in PLA/MWCNT composites has been proven in bibliography. PLA/MWCNT composites display easier processability by 3Dprinting and more adhesion between the printed layers whit lignin than with PEG’.

Line 99:  ‘The use of lignin as a bio-plasticizer was compared with PEG, a synthetic and well-known additive for the PLA/MWCNT composite.’

In this case, we wanted the additives (lignin or PEG) to act as MWCNT dispersants to obtain MWCNT agglomerates well-interconnected through single MWCNT. This microstructure favors the electrical conductivity in polymer matrices.  The functionalization of MWCNTs produces greater compatibility with the PLA matrix, improving the distribution and dispersion of MWCNT and as consequence, the CNT clusters tend to break down and the conductivity probably decreases.

6-Introduction:

  • 1 The introduction is written very briefly. Also, the sources reviewed in the introduction are very few. The introduction should be completely rewritten. The introduction of FDM, pure thermoplastic and composites, should be added in the introduction.

We thanks for the work and effort that the reviewer has done. We have rewritten the introduction accordingly to the referee comments:

  • Introduction of FDM (line 33): ‘The Additive Manufacturing (AM)[1,2] industrial revolution is very promising regarding the fabrication of electronic devices[3] due to its many advantages (no waste production, final product adaptability, low cost, etc). One of the most important [4–7] AM techniques is Fusion Deposition Modelling (FDM) [8], basing its operation on the melting of a thermoplastic material [9].’
  • Pure thermoplastics: (line 38) ‘Nowadays there are commercially available thermoplastic filaments to easily feed the 3D printer (acrylonitrile butadiene styrene (ABS), polyethylene terephthalate glycol (PETG) or polylactic acid (PLA). Moreover, researchers are looking for novel filaments from other thermoplastic polymers such as PVC [10] or PLA-TPU [11]. However, these materials possess a lack of functional properties (such as electrical conductivity) to be used in the fabrication of such electronic devices. Readily accessible filaments with functional properties such us electrical conductivity exist on the market but they are still scarce and expensive, so FDM applications in the electronic field are limited [12].’
  • Composites (line 48): ‘The development of conducting polymer composites (CPC), where a conductive filler is incorporated into a thermoplastic matrix, is a good alternative for the production of printable conductive filaments [8,12,13], whenever materials should display suitable physical properties (electric, rheological, thermal and mechanical properties) to 3D printing process. Previously published researches display the used of conductive polymer composites in energy storage devices [6,14], such as electrodes [15–17] or electrolytes [18], in electromagnetic interference shielding [19,20], electronics industry [21] and biomedical application [22,23].’

In addition, we have completed the paragraph before Table 1 to highlight the novelty of our work (line 86): ‘In the current work, we propose a combination of two methods (solution and melt mixing) to obtain a highly electrically conductive material with easy printability. This new composite will allow all the advantages of the technique to be exploited in the fabrication of new products for electronic applications.’ With this changes and rearrangement, we believe that now the introduction is more understandable.

6.2-It is suggested to use these references which are for new materials such as PVC, PLA-TPU and PETG (“Development of Pure Poly Vinyl Chloride (PVC) with Excellent 3D Printability and Macro‐and Micro‐Structural Properties” --- “3D printing of PLA-TPU with different component ratios: Fracture toughness, mechanical properties, and morphology”).

The author thanks the referee for the observation. We think that it is enriching to add the new 3D printable materials to this introduction and that is the reason why we have added the following paragraph with the references that the reviewer suggested:

Line (40): ‘Researchers are looking for novel filaments from other thermoplastic polymers such as PVC [10] or PLA-TPU [11]’.

  • 3-Why is the melt mixing method not used directly in the internal mixer or extruder?

The author thanks the referee for the commentary. When we were planning this investigation, we decided to combine solvent-casting with melt mixing for various reasons:

  1. Handling nanoparticles directly in the extruder can be hazardous to health, as they can be inhaled by remaining in the air. So we decided to add MWCNT to PLA by solvent-casting in an extract-hood with a protection mask, in a controlled environment.
  2. Thanks to the solvent-casting first step, MWCNT have a first dispersion through the PLA, this step enables a higher electrical conductivity in the end of the process.

To better explain this in our work, we added a sentence in the introduction:

Line 95: The obtaining of composite was optimized by a combination of solvent-casting and melt-mixing, getting highly conductive polymer composites in a safe way.’

  • 4-It is suggested to summarize the construction parameters in a table.

Surely the addition of a new table would make the text more reader-friendly, but given the number of tables that already exist and that such data is already expressed in the text, we consider that it is not necessary.  In addition, we have improved the injection parameters explanation (line 130-132):  ‘PLA/MWCNT composite were injection-molded in a Haake MiniJet Pro (Thermo Scientific) at 210 oC (mold at 60oC), a pressure of800 bar during 6 seconds and post-pressure of 500 bar during 3 seconds to evaluate their physical properties’

However we leave it to the editor's discretion whether or not to incorporate the table attached below:

Processing technique

Temperature [oC]

Speed [rpm]

Time

[s]

Pressure [bar]

Post-pressure [bar]

Extrusion

210

40

300

-

-

Injection

210 and 60

-

6 and 3

800

500

Besides, the 2.2 3D Printing process section was added to reorganize the information about the printer’s parameters.

Line 144:

2.2. 3D printing process

PxCyL and PxCzG composites were shaped into coins by 3D printing process. The selected 3d printer was a modified creality CR-10 v2 with a pellet extruder. The printing conditions were 200 oC at the extruder, hot bed at 60 oC, nozzle of 0.8mm, layer high of 0.3mm, superficial ironing of 5% flow rate with 0.1 mm of separation between ironing passes and printing speed of 10 mm∙s-1. The samples were printed with 100% infill, lineal pattern and 3 external perimeters.

  • 5-Composites in a table with the weight percentage of nanoparticles and polyethene glycol and lignin should be presented.

The author thanks the referee for the note. PLA/MWCNT with additives (lignin or PEG) formulations are collected at the Table 4. Since this table is at the very last of the paper, a reference to this table was added in the beginning of the work, at ‘Preparation of the composites’, second paragraph.

Line 126: ‘Then, the PLA/MWCNT (Table 3), PLA/MWCNT/Lignin and PLA/MWCNT/PEG (Table 4) composites were melt-blending’

  • 6 How has the reproducibility of the results been checked?

The author thanks the referee for the question. Every characterization was repeated to ensure reproducibility:

SEM (line 155): ‘At least 2 samples of each formulation were broken, prepare and analyze by SEM to en-sure an acceptable reproducibility.’

Electrical conductivity (line 162): ‘The σ obtained for each composite formulation are the result of at least 12 measurements on three different samples on the top and bottom surfaces to confirm the homogeneity of the circular plaques.’

Rheology (line 174): ‘Each curve reported is an average of at least two samples.’

Tensile test (line 182): ‘At least five specimens of each sample were tested to obtain the average value of the mechanical properties and their standard deviations.’

DSC (line 192): ‘Every result is the average of at least 2 measurements in different specimens.’

  • 7 The print section of the samples has been forgotten. It is necessary to provide this section due to the effect of printing parameters.

We appreciate the reviewer’s proposal. A new section regarding the 3D printing process has been included:

Line 144:  ‘PxCyL and PxCzG composites were shaped into coins by 3D printing process. The selected 3d printer was a modified creality CR-10 v2 with a pellet extruder. The printing conditions were 200 oC at the extruder, hot bed at 60 oC, nozzle of 0.8mm, layer high of 0.3mm, superficial ironing of 5% flow rate with 0.1 mm of separation between ironing passes and printing speed of 10 mm∙s-1.

7-Results section:

  • 1 Is it mandatory to submit section 3.1?

We appreciate the reviewer’s proposal. However, we believe that section 3.1 is necessary since this work is based on the combination of solvent-casting and melt-mixing techniques so the search of the best solvent, or in this case solvents combination, has to be part of the work. Moreover, we decided to run this study to get a solvent combination which damage the less the PLA matrix. In addition, part of the novelty of this work is precisely the combination of the two techniques.

  • 2 The parameters of Table 2 should be introduced.

The author thanks the referee’s advice. This point has been corrected:

Line 214: ‘Table 2. Solvent optimization study samples composition and tensile strength results. (Where E refers to young modulus, σB is the stress at the break point and εB the strain at the break point).’

  • 3 How are the mechanical properties checked?

The author thanks the referee for the question.  The mechanical properties were measured by tensile test, using an Instron 5566 universal. To clarify this in the ‘characterization methods’, section 2.2.4, we have changed a bit the first sentence:

Line 177: ‘The mechanical properties were measured by uniaxial tensile tests, performed at a crosshead speed of 2 mm∙min-1 at room temperature using an Instron 5566 universal test machine (Instron, Canton, MA) according to UNE-EN ISO 527-2.’

  • 4 The SEM images used are raw. These images should be corrected. Add a scale bar to them. By adding text to these images, they can be used for better analysis of the results.

The author thanks the referee for the observation. However, we didn’t put the SEM images raw, they were treated, selected from a lot of micrographs to ensure reproducibility and the important zones of each image was zoomed or marked as convenient. In addition, every micrographs possess its scale bar (in the bottom of each image). We have included to every image the formulation code, to make it more reader friendly, as it was suggested by the reviewer:

  • 5 Are these images for printed samples?

We appreciate this observation. To clarify how were the samples obtained, the legends of Figures 3 and 6 were modified:

Line 270: ‘Figure 3. SEM micrographs of injected samples of PLA with (A) 2, (B) 4, (C) 5 and (D) 10 percent of MWCNT, amplitude x40 and inset x5000.’

Line 409: ‘Figure 6. SEM micrographs of 3D printed samples of PLA/MWCNT with (A) 1, (B) 2 and (C) 3 percent of Lignin, (D) 1 (E) 2 (F) 3 percent of PEG.’

  • 6 The results of the physical properties of the printed samples are presented in what plane and raster angle?

The author thanks the referee for the commentary. Printed samples were evaluated by SEM, rheology and electrical conductivity. Rheology wasn’t affected by the printing since the samples were melted. SEM was run at the cross-section fracture in both injected and 3D-printed samples, as it is now explained at 2.2.1 section: ‘The fracture was made to evaluate the cross-sectional microstructure of the samples. At least 2 samples of each formulation were broken, prepare and analyze by SEM to ensure an acceptable reproducibility.

Regarding electrical conductivity measurements, it was run at their top (were a superficial ironing was made) and in the bottom (in which the sample has been in direct contact with the printing bed). In both surfaces the measurements were made at least in 4 different directions to ensure homogeneity of the measurement:  Section 2.2.2 ‘The σ obtained for each composite formulation are the result of at least 12 measurements on three different samples on the top and bottom surfaces to confirm the homogeneity of the circular plaques.’

  • 7 Has the effect of printing parameters been checked? For example, number of microholes caused by the printing strongly affect the electrical properties?

The authors recognize the referee’s commentary. In FDM 3D printing there are a lot of parameters that can be tuned or modified to improve the quality of the printing (such as the temperature of the extruder and of the print bed, speed of the extruder, nozzle size, layer high, retraction speed, superficial ironing, etc.) and probably all of them interfere in the final electrical properties of the samples. Despite we think that this study could be very interesting and beneficial to our work, we did not include it in the current work since it would require a lot of new experiments. That is why we have decided to make a separate study to evaluate the 3D-printer parameters influence on the final electrical conductivity of the samples.

  • 8 Why is the CNT distribution not studied for the printed samples?

The author thanks the referee for the estimation. The MWCNT distribution and dispersion through the PLA was studied by SEM and by electrical conductivity measurements on 3D-printed samples. The study of the nanocomposites’ morphology by another technique such as TEM would give more information on the dispersion of the individual CNTs, however, as discussed above in the answer to the question 5, the more efficient conducting network is formed by CNT clusters well-interconnected by single CNTs. This structure is more visible by SEM than by TEM and for this reason, the last one was chosen.

  • 9 How is the accuracy of electrical property results checked?

The author thanks the referee hard work doing this positive revision. To check the accuracy of the electrical properties, we used an ESP probe checker (0.996 Ω) before every set of measurements. In addition, as it is explained in 2.2.2 section:

Line 162: ‘The σ obtained for each composite formulation are the result of at least 12 measurements on three different samples on the top and bottom surfaces to confirm the homogeneity of the circular plaques.’

Reviewer 3 Report

Dear,

The authors produced conductive PLA/MWNCT compounds through 3D printing. The versatility for molding polymeric products on a 3D printer is on the rise. The manuscript is a good contribution to the literature.

> Introduction. Please make clear the gap in the literature on the topic and the novelty of the manuscript;

> Why didn't the authors add a detailed characterization of the lignin used?

> The authors should discuss aspects of processing in the formation of filaments. For example, it is usual to perform a surface analysis by optical microscopy to check roughness;

> Figure 3. Transmission microscopy (TEM) analysis would be more appropriate, considering the nanometric scale of carbon nanotubes;

Author Response

Referee 3: The authors produced conductive PLA/MWNCT compounds through 3D printing. The versatility for molding polymeric products on a 3D printer is on the rise. The manuscript is a good contribution to the literature.

  1. Please make clear the gap in the literature on the topic and the novelty of the manuscript.

The author thanks the reviewer appreciation. To highlight the novelty of our work, we have added a paragraph in the introduction:

Line 85: ‘In the current work, we propose a combination of two methods (solution and melt mixing) to obtain a highly electrical conductive material which can be process by easily 3D printing, with all the advantages of the technique’.

In addition, the last paragraph of the introduction makes a summary of the main points of the current work. We believe that this paragraph focus the reader on our investigation. (Lines 93-103)

  1. Why didn't the authors add a detailed characterization of the lignin used?

We thanks the referee for the comment. We had forgotten to add a reference here from our collaborator that provide us with lignin. This paper explains perfectly the obtaining process of the lignin used. In addition, we added two sentences with to the ‘materials and methods’ section in which the main characterization is briefly explained:  

Line 112: The obtained lignin was extracted from acetic acid solution by water precipitation and lyophilized before use. Its characterization was done by gel permeation chromatography, obtaining Mw=2800, Mn=1706 and polydispersity = 1.6 (average of 2 replicates)’.

  1. The authors should discuss aspects of processing in the formation of filaments. For example, it is usual to perform a surface analysis by optical microscopy to check roughness.

The author appreciates the referee positive feedback. However, in this study we tried to focus on the evaluation of lignin as bio-additive of PLA/MWCNT composites, testing its effect mainly on the electrical conductivity and printability by FDM. That is why we focused on the improvement of the materials’ formulation, since the optimization of the best solvent formulation dissolve the PLA without damaging it, to the electrical threshold of PLA/MWCNT and its whole characterization to finally adding lignin and evaluating it. Despite this feedback could be very enriching to our work, we have decided to use it in a different in depth future study.

  1. Figure 3. Transmission microscopy (TEM) analysis would be more appropriate, considering the nanometric scale of carbon nanotubes.

The author thanks the referee for the comment and for its work and effort reviewing this article. (This question was answering in part in point 7.8 of reviewer 2) Our samples are very heterogeneous, and the needed sample for TEM analysis would be that small that results would not be representative. What we are trying to achieve in this work is an enhancement of PLA/MWCNT composite electrical conductivity and 3D printability. Therefore, by SEM we study lignin effect on the distribution and dispersion of the nanofillers. However, for the purpose of this work it more important to see MWCNT agglomerates than the dispersion of single MWCNT. For that reason, we decided to not use TEM and use SEM instead. 

Round 2

Reviewer 1 Report

accept as it is

Reviewer 2 Report

Accept.